# Molecular Impact of Conventional and Electronic Cigarettes on Pulmonary Surfactant

**DOI:** 10.3390/ijms241411702

**Published:** 2023-07-20

**Authors:** Maria Lisa Garavaglia, Francesca Bodega, Cristina Porta, Aldo Milzani, Chiara Sironi, Isabella Dalle-Donne

**Affiliations:** 1Dipartimento di Bioscienze, Università degli Studi di Milano, 20133 Milan, Italy; aldo.milzani@unimi.it (A.M.); isabella.dalledonne@unimi.it (I.D.-D.); 2Dipartimento di Fisiopatologia Medico-Chirurgica e dei Trapianti, Università degli Studi di Milano, 20133 Milan, Italy; francesca.bodega@unimi.it (F.B.); cristina.porta@unimi.it (C.P.); chiara.sironi@unimi.it (C.S.)

**Keywords:** conventional cigarette smoke, electronic cigarette vapor, lung, alveolar surfactant, surfactant lipids, surfactant proteins

## Abstract

The alveolar epithelium is covered by a non-cellular layer consisting of an aqueous hypophase topped by pulmonary surfactant, a lipo-protein mixture with surface-active properties. Exposure to cigarette smoke (CS) affects lung physiology and is linked to the development of several diseases. The macroscopic effects of CS are determined by several types of cell and molecular dysfunction, which, among other consequences, lead to surfactant alterations. The purpose of this review is to summarize the published studies aimed at uncovering the effects of CS on both the lipid and protein constituents of surfactant, discussing the molecular mechanisms involved in surfactant homeostasis that are altered by CS. Although surfactant homeostasis has been the topic of several studies and some molecular pathways can be deduced from an analysis of the literature, it remains evident that many aspects of the mechanisms of action of CS on surfactant homeostasis deserve further investigation.

## 1. Introduction

The large parenchymal area of the lung, needed to ensure gaseous exchange, also maximizes the deleterious effects of environmental, physicochemical or microorganism-mediated stressors.

Cigarette smoke (CS) has a more detrimental effect on the lungs than other pollutants. Indeed, it is inhaled through the oral cavity, completely bypassing the filter of the nasal cavities, and reaches all the organs of the respiratory system at concentrations higher than any other environmental contaminant, leading to respiratory diseases such as interstitial lung disease, emphysema, cancer and chronic obstructive pulmonary disease (COPD)—one of the most common preventable diseases among smokers, characterized by persistent and generally progressive airflow limitation, associated with an increased chronic inflammatory response in the airways and the lungs caused by exposure to noxious particles or gases [1]—common in both aged and young smokers [2].

Alveolar flow and particle deposition are not yet fully understood, because of the complex geometrical structure, limited accessibility and variable geometry of the alveolar region during breathing [3,4]. Despite this, it is known that several noxious CS components—especially the most lipophilic gaseous components—can penetrate to the alveoli, with deleterious effects on the alveolar parenchyma, affecting its morphology and functionality [5].

The alveolar parenchyma consists of two superimposed thin cell layers constituting a structural barrier separating the air from the blood: a simple epithelial sheet and an endothelium, facing the alveolar and the capillary lumen, respectively. Between them occurs an interstitial space of different thickness and composition, comprising other cell types, such as fibroblasts, nervous cells, lymphoid cells, macrophages and dendritic cells [6].

The alveolar epithelium—constituted by flattened gas-exchanging alveolar type I cells (ATI) and secreting alveolar type II cells (ATII)—is covered by a non-cellular layer, with an average thickness of 0.2 µm [7], constituted by a tiny aqueous hypophase topped by pulmonary surfactant (Figure 1), a complex lipo-protein mixture with surface-active properties [8].

The surfactant consists of approximately 90% lipids, mostly saturated phospholipids, and surfactant proteins (SP-A, B, C and D). The surfactant components are synthesized in the alveoli by the ATII cells, in which the hydrophobic fractions of the surfactant (SP-B, SP-C and lipids) are assembled to form specific organelles, the multilamellar bodies (MLBs), where the surfactant is stored before secretion [9,10]. The secretion of MLBs in the aqueous hypophase occurs mainly in response to mechanical stimuli (inflation) [9,11]. Furthermore, the surfactant is produced by the Clara cells, secretory cells located between the ciliated cells of the terminal bronchiole epithelium, where the surfactant is involved in maintaining small airway patency and in transporting particulate from the alveolar surface to the small airways [12].

The surfactant settles at the air–liquid interface in the alveolar space after being modeled into intermediate transient forms, decreasing the surface tension of the fluid that covers the alveoli to a very low value. As a result, the surfactant reduces the amount of mechanical work required to expand the alveolar surface during inhalation and prevents alveolar collapse. The ability of the surfactant to lower the surface tension is surface-area-dependent, increasing as the alveoli become smaller, thereby contributing to alveolar stability. The low surface tension at the alveolar air–liquid interface also helps to prevent the formation of intra-alveolar edema [8,13,14]. Besides this purely biophysical relevance, several proteins [15] or lipid molecules [16] of the surfactant have immunomodulatory functions.

Once secreted and settled, the exhausted surfactant is absorbed, degraded or recycled by ATII cells or by the macrophages populating the aqueous hypophase [9].

The non-cellular layer beneath the surfactant comprises the glycocalyx of the epithelial cells, consisting of glycoproteins and proteoglycans anchored to the epithelial membrane. These molecules form a highly hydrated molecular mesh of modulable viscosity and thickness, constituting a medium that regulates pulmonary functions such as gaseous exchange, the immunomodulatory activity of surfactant proteins, the alveolar macrophage function and the alveolar liquid flow that removes particles from the alveoli [17]. Besides binding numerous water molecules, the glycocalyx’s constituents can link various molecules through ionic and/or hydrophobic non-covalent bonds, interacting with inflammatory cells, proteases and growth factors, which also play a fundamental role in the development and repair of tissue and in the onset of numerous pathologies [18].

The maintenance of the pulmonary fluid layer is accomplished through the concerted regulation of several transporters and ion channels present in the ATII and ATI cells [19,20,21]. Ionic transporters, by regulating, for example, the pH and Ca^2+^ concentration of the hypophase or its water content, influence the biophysical properties and activity of extracellular surfactant.

Several studies suggest that CS has deleterious effects on the hypophase and surfactant homeostasis, rheological characteristics and functions, but many aspects of the impact of CS on surfactant composition and metabolism are still incompletely understood. The aim of this review is to provide an overview of the most recent findings, analyzing the possible mechanisms underlying the molecular alterations caused by CS and their possible correlation with the development of lung diseases. Moreover, the diagnostic and/or prognostic potential of SP alterations in cigarette smokers’ serum is addressed.

## 2. Lipid and Protein Composition of Surfactant

The surfactant is constituted by a complex mixture of lipids and proteins whose composition has to be finely regulated to achieve the surface-active properties necessary to ensure proper lung functionality. Numerous excellent reviews on this topic are available [22,23]. Here, we briefly summarize some information about the composition and metabolism of surfactant lipids and proteins, important for a better understanding and analysis of CS’ effects.

Lipids constitute the largest proportion (approximately 90% by weight) of the pulmonary surfactant. The composition and the metabolism of lipids in alveolar surfactant were extensively analyzed in the second half of the twentieth century, and their alterations are still broadly studied, particularly in view of their possible pathological implications [24,25].

The amphipathic nature of the surfactant is mainly attributable to the presence of the zwitterionic dipalmitoyl-phosphatidylcholine (DPPC), a saturated C16 diacyl phospholipid, which is responsible for the reduction of the alveolar surface tension, as it can pack into highly compact condensed films at the air–liquid interface.

Other surfactant phospholipids contribute to surfactant fluidity, surface adsorption and biological or immunological activity. In fact, the content of phosphatidylcholine appears to be preserved in different species, whereas the content of other components varies considerably, according to the different environmental conditions in which the organisms live [26,27].

DPPC, together with palmitoyl-myristoyl-PC (PC16:0/14:0) and unsaturated palmitoyl-palmitoleoyl-PC (PC16:0/16:1), constitutes the predominant lipid fraction (75–80%) of mammalian surfactant. The proportions of these cationic phospholipids can vary during development or in some pathological conditions [28], promoting subsequent pathogen infections, as recently demonstrated for COVID-19 in obese patients [29].

The presence of unsaturated lipids and cholesterol allows the creation of lipid domains with higher fluidity, which are physiologically important as they make the surfactant adaptable to the dynamic variation in the alveolar surface during respiration. Indeed, they modulate surfactant absorption and re-spreading during the respiratory cycle [30].

Phospholipids with unsaturated acyl side chains, which represent approximately 30% of the surfactant phospholipids, are more susceptible to the oxidizing action of environmental factors or to the activity of pro-oxidant enzymes such as lipoxygenase, myeloperoxidase, or NADPH oxidase. Oxidized phospholipids, such as 1-palmitoyl-2-(5–oxovaleroyl)-sn-glycero-3-phosphocholine, malondialdehyde (MDA) and 4-hydroxynonenal (HNE), in addition to modulating the immune response, have biological and biophysical properties that differ from those of their precursors. They can be related to the development of different diseases, including in the lungs [31,32].

Cholesterol, as with unsaturated phospholipids, can undergo oxidation either in the ring structure or on its side chain, generating bioactive lipids involved in the development of inflammation [31]. Although high cholesterol in surfactant has been correlated with several respiratory diseases and surfactant dysfunction [33,34,35], it has recently been demonstrated that its presence is important for proper interaction with the oligomeric complex of SP-B [36].

Phosphatidylglycerol (PG; mainly palmitoyl-oleoyl-phosphatidylglycerol—POPG) is the anionic phospholipid present in the greatest amount in surfactant (8% by weight). It modulates macrophage function [37] and, together with other anionic phospholipids, such as dioleoyl-phosphatidylinositol (PI, 3% by weight), regulates the innate immune processes mediated by toll-like receptors, and it is a powerful anti-viral agent for the respiratory syncytial virus, influenza A and SARS-CoV-2 viruses [22]. PG and PI, establishing ionic interactions with the hydrophobic surfactant proteins (SP-B and SP-C) [38], regulate the surfactant’s properties by stabilizing its structure during the dynamic phases of the respiratory cycle [39]. Furthermore, by interacting with SP-D, they facilitate the reuptake of the surfactant by the ATII cells [25].

Another phospholipid present in small quantities (1.5% by weight) is phosphatidylethanolamine (PE), which, by interacting with the hydrophobic proteins SP-B and SP-C and because of its peculiar structural characteristics, makes energetically favorable the adoption of a negative curvature by the lipid bilayer. This is important in respiratory processes and during the adsorption of the vesicular bilayer of the MLBs to the air–water interface [40]. Furthermore, PE has an antifibrotic effect as it inhibits the production of collagen by normal human lung fibroblasts [41].

Cardiolipin is a lipid present in smaller quantities than others but has important roles in lung surfactant homeostasis [42].

Plasmalogens constitute a low proportion of surfactant and have antioxidant and structural properties, contributing to the reduction of the surface tension at the alveolar surface [43].

Other lipid molecules recently identified consequently to the increasing use of mass spectrometry are ether phospholipids, in which one of the two side acyl chains (sn-1 position) is linked to the glycerol backbone with an ether instead of an ester bond [44]. Although the function of the ether phospholipids in the cellular physiopathology has been at least partially characterized [45], their function in surfactant remains unknown; however, their level significantly correlates with the forced expiratory volume in 1 s (FEV1), suggesting a role in regulating pulmonary compliance via surfactant [44].

Sphingolipids (particularly sphingomyelin) have been recently identified as constituents of the surfactant, and their level is positively correlated with FEV1 [44].

Ceramides, sphingolipids in which a molecule of N-acetylsphingosine is linked to a variable-chain fatty acid, are present in the surfactant at rather variable levels in different individuals [44]. The degradation of sphingomyelin to ceramide, mediated by some cytokines, such as TNFα, appears to participate in some pathological processes related to acute lung injury. Ceramides impair the biophysical properties of the alveolar surfactant film, reducing its surface activity [46]. A marked increase in sphingolipids has been demonstrated in pulmonary alveolar proteinosis [47] and in pathological hyperinflammation [16].

Lysobisphosphatidic acid (or bis(monoacylglycerol)phosphate—BPM) is synthesized from phosphatidylglycerol by alveolar macrophages [48]. Its function is not yet known, but its levels correlate with pulmonary FEV1 [44]. Interestingly, its increase has been observed in mucopolysaccharidosis IIIA, a lysosomal storage disease characterized not only by skeletal and neurological alterations but also by respiratory dysfunction [49].

The polymorphism of surfactant lipids, their energetically favorable structural conformation and their subsequent impact on the uptake and organization of the surfactant lipid and adaptability to compression are described in a comprehensive recent review [27].

The pulmonary epithelium has high lipogenic activity, maintained by transport and enzymatic mechanisms that are not different from those of other cell types, because of the continuous synthesis and turnover of surfactant [50]. Exhaustive recent reviews illustrate the transport and enzymatic mechanisms involved in lipid homeostasis in the lung [24,51,52], the alteration of which is related to the development of pulmonary diseases.

Besides lipids, surfactant includes proteins (approximately 10% by weight). Among these, specific proteins (6–8% by weight) are essential to the surfactant’s structure and function [10,53] and comprise both the hydrophilic SP-A and SP-D and the hydrophobic SP-B and SP-C, with the first one being the most abundant lung-specific protein [53,54].

Proper homeostasis of all surfactant components is essential for its functionality, requiring strict lipid–protein and protein–protein interactions [54,55,56,57].

The four specific proteins exhibit markedly different structures, functions and biochemical properties.

SP-B and SP-C are present in a variety of other human tissues and fluids besides the lung, including the testis, gingiva, salivary glands and saliva, nasolacrimal and ocular surface tissues and tear fluid [58,59,60]. They are small, highly hydrophobic cationic proteins, assembled in ATII cells with lipids to form MLBs, the proper formation of which strictly requires the presence of SP-B [10,61,62]. Once secreted, MLB membranes unpack and are adsorbed at the air-liquid interface, forming the functional surfactant film, with a process highly dependent on the specific lipid-protein composition. DPPC alone cannot be efficiently adsorbed at the air–liquid interface, and the presence of SP-B and SP-C, in addition to anionic and unsaturated phospholipids, is therefore crucial for this task [36,63,64,65].

SP-B is a member of the saposin or saposin-like protein (SAPLIP) family of proteins, which have important impacts on phospholipid organization [36,66,67]. Synthesized as a proprotein (proSP-B), SP-B is proteolytically processed to an 8-kD protein that associates to form sulfhydryl-dependent ~18 kD homodimers [68,69]. The 381-amino-acid pro-SP-B is processed sequentially into the mature SP-B form with the involvement of the cleavage activity of several intracellular enzymes, such as napsin A, cathepsin H and pepsinogen C [70]. As with other members of the saposin-like protein family, it contains six highly conserved cysteines that form intramolecular disulfide bonds. SP-B has a seventh cysteine that forms the intermolecular disulfide bond of the homodimer.

Mature SP-B has an amphipathic α-helical secondary structure flanked by apolar loops. It strongly associates with the acyl chains of phospholipids, permitting the orientation of the protein parallel to the plasma membrane surface [71,72]. SP-B, having a positive net charge, interacts preferentially with anionic phospholipids, such as PG [67,73].

SP-C is the smallest (4.2 kDa) and most hydrophobic surfactant protein, with a 35-amino-acid protein sequence, arranged as an α-helix spanning between aminoacidic positions 9 and 34 and with a transmembrane orientation [74]. SP-C is considered a support molecule since its functions highly overlap with those of SP-B in accomplishing the complex and dynamic physiology of the surfactant [57]. The N-terminal portion of SP-C has an amphipathic character with a net positive charge and two palmitoylated cysteines, which permits it to increase the interaction of SP-C with anionic phospholipids such as PG. Furthermore, both palmitic chains, interacting with the acyl side chains of lipids, anchor the protein to the lipid membranes, preventing its exclusion from the highly compressed surfactant lipid films [75].

Besides promoting surfactant adsorption at the alveolar air–liquid interface, SP-B and SP-C provide lipid film stability during respiratory dynamics. Indeed, they regulate the surface lateral diffusion of phospholipids, collecting surfactant material squeezed out during exhalation (compression) and re-spreading it upon inspiration (expansion). A multi-layered surfactant reservoir is thus generated, which also stores newly secreted surfactant complexes [10,14,36,57,64,76,77]. Finally, SP-C transfers lipids among different surfactant lipid structures and inhibits the degradation of surfactant phospholipids by plasma proteins [57,69].

Although hydrophobic proteins are major players in surfactant homeostasis, the SP-B/SP-C machinery requires the presence of the hydrophilic protein SP-A, which enhances the adsorption of surfactant phospholipids at the alveolar air–liquid interface and participates in the rearrangement of the surface film during compression–expansion respiratory cycles [10,56]. Moreover, SP-A inhibits surfactant secretion [78,79], in opposition to SP-B, which is a secretion inducer [63]. Finally, SP-A promotes exhausted surfactant reuptake by ATII cells [10,54,79]. This process requires prior surfactant fragmentation into small vesicles that is mediated not only by SP-A but also by SP-D, which plays a role in the homeostasis of the surfactant pool size [10,80].

SP-A and SP-D are synthesized in the endoplasmic reticulum of ATII cells and secreted via vesicular transport, although a minimal amount of SP-A should be directed to MLBs [9,81].

These hydrophilic proteins share extensive structural similarities and belong to the “collectin” (collagen-lectin) family, whose members are characterized by the presence of multiple globular “head” regions connected by collagen-like triple helix strands [81,82]. SP-A has binding domains for phospholipids, glycolipids, lipopolysaccharides, carbohydrates and calcium ions [73] and, as with SP-D and all collectins, can bind the C1q receptor expressed by many cell types, including macrophages, acting as opsonins [81,83,84].

The alveolar lining fluid is important in lung innate immune defense, being the first line of host protection against inhaled harmful material, not only as a physical barrier but also via the two surfactant collectins, which can bind allergens and pathogens, thus enhancing their clearance. Furthermore, SP-D and SP-A show many antimicrobial effects, preventing the dissemination of infectious agents [81,84,85].

In fact, SP-A and SP-D (i) show bacteriostatic effects, directly affecting the proliferation and viability of microorganisms, inhibiting the growth of various bacteria and increasing their membrane permeability [10,86]; (ii) can bind directly to a broad range of microorganisms, thanks to the ability to recognize several pathogen-associated molecular patterns (PAMPs) on their surfaces [86,87,88]; (iii) prevent pathogen adhesion to alveolar epithelial cell surfaces and the consequent microbial internalization, colonization and invasion, thus reducing infectivity [10]; (iv) determine microorganism agglutination, therefore facilitating their mucociliary clearance [86]; (v) through simultaneous binding to bacteria and neutrophil extracellular traps (NETs), favor bacterial trapping, killing and clearance [89]; (vi) acting similarly to opsonins, stimulate alveolar macrophages’ ability to uptake and destroy viruses, bacteria, mycobacteria, fungi, allergens and apoptotic cells [25]. Finally, being able to bind several classes of immunoglobulins, SP-D represents a connection between the adaptive and innate immune systems [90].

Besides antimicrobial effects, SP-A and SP-D contribute to alveolar surface defense and homeostasis, regulating inflammation [83]. A balance is required between inflammatory responses to defend the lung and the need to prevent exacerbated reactions because of the huge number of external noxious particles and microorganisms to which the lung surface is exposed [83]. SP-A and SP-D modulate the cellular inflammatory response in a context-dependent fashion: in the presence of infectious agents, these proteins increase pro-inflammatory mediator release, whereas, in physiological conditions, they limit the production of soluble mediators and the activation of inflammatory processes both in macrophages and epithelial cells to avoid a chronic inflammatory state [25,91].

Finally, SP-A and SP-D influence alveolar epithelium homeostasis by regulating the apoptosis of epithelial cells. Indeed, SP-D inhibits apoptosis, both collectins enhance the phagocytic clearance of apoptotic cells by alveolar macrophages, and SP-A promotes alveolar cell proliferation and lung epithelium renewal and repair [10,84,85].

SP-D regulates ceramide species balance, lowering the levels of C16 and C24 ceramides. These are pro-inflammatory, pro-apoptotic and pro-autophagic bioactive sphingolipids, which are increased after exposure to CS in vivo, leading to alveolar epithelium damage, lung injury and COPD pathogenesis [92].

High endogenous SP-D levels reduce COPD risk and lung function impairment, suggesting SP-D as a therapeutic target [92,93].

In summary, SPs have been shown to provide immune protection against respiratory pathogens, both directly by limiting inflammation and promoting pathogen clearance, and indirectly by activating molecular and cellular mechanisms that contribute to restoring lung homeostasis [10].

## 3. Effect of Cigarette Smoke on the Lipid Components of Surfactant

### 3.1. Cigarette Smoke’s Effect on Composition and Abundance of Surfactant Lipids

In this overview, only the results of analyses conducted on healthy subjects and wild-type animal models are included, to highlight the effects exclusively attributable to exposure to CS. Furthermore, to achieve a complete outline of the effects, sometimes contradictory, of CS on the lipid components of surfactant, we include all the studies conducted since the 1970s to the present.

To facilitate the identification and comparison of the many lipids that compose surfactant, the comprehensive classification of lipids proposed by Fahy and colleagues [94] is followed, dividing lipids into six categories (fatty acids, glycerolipids, glycerophospholipids, sphingolipids, sterol lipids and prenol lipids) comprising distinct classes and subclasses of molecules (Table 1 and Table 2).

Most human studies have been conducted by analyzing the lipid fraction of bronchoalveolar lavage (BAL). The appropriateness of BAL to assess the components of the non-cellular layer lining the alveoli remains uncertain, as it is extremely difficult to determine what percentage of the fluid recovered from BAL represents the alveolar fluid from distal airways and alveoli [95].

In a pioneering paper published in 1972 [96], a marked reduction in surfactant lipid levels in smokers compared to non-smokers was observed. This reduction returned rapidly to non-smoker levels after smoking cessation.

Regarding the lipid composition of the surfactant, some lipids appear to be dysregulated, with heterogeneous results. However, in the majority of studies, the concentration of phosphatidylcholine remains constant after exposure to CS (Table 1). The latter observation is in agreement with the previous findings [97], where a change in the non-polar lipid fraction but not in the phospholipid levels was observed, and with more recent findings [98,99]. Conversely, another study reported [100] that the total quantity and the concentration of phosphatidylcholine in BAL were lower in light smokers than in non-smokers, in agreement with the reduction observed in a previous study [96], and were inversely related to the number of cigarettes smoked.

The concentration of other phospholipids is also significantly dysregulated. A decrease in the concentration of phospholipids was demonstrated in smokers’ BAL fluid [101] and CS was proven to alter the lipid composition of surfactant [102], inducing a selective increase in the concentrations of phosphatidylethanolamine, sphingomyelin and phosphatidylglycerol, but not of phosphatidylcholine. An increase in phosphatidylethanolamine was observed elsewhere [103].

The concentration of neutral lipids also appears to be increased or reduced in smokers, particularly that of cholesterol, which was detected to be decreased in past studies [96,102], conversely to what was observed in a more recent study conducted on the induced sputum supernatant [99].

In conclusion, the analysis conducted on humans (Table 1) shows rather variable findings. The results, which sometimes appear discordant, could be attributable to the different smoking habits of the subjects under examination, not always defined; to individual variability, also influenced by the limited number of subjects examined; to age, a variable known to influence surfactant composition [104]; or to methodological experimental variability, such as the lipid extraction or the BAL recovery method.

The situation therefore remains unclear and deserves to be further investigated, mostly in consideration of the development and increasing accessibility of methods such as mass spectrometry, which could easily clarify this topic.

**Table 1 ijms-24-11702-t001:** Dysregulation of different lipid types or classes of alveolar surfactant in smokers vs. non-smokers.

Subjects;Smoking Habit	Analysis	Tot.Lipids	FA	GL	Glycerophospholipids	Sphingolipids	Sterols	Prenols	Ref.
				Pa	PC	PG	APG	PE	PI	PS	SM	Cho	CE		
Never smokers: 8, mean age 26 years.Current smokers: 8, mean age 24 years.*HS*	Acellular BAL	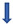	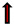	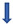		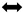			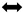	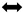	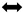	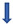	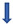	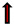		[96]
Never smokers: 12, mean age 29.9 years.Current smokers: 14, mean age 28.5 years.*ND*	Acellular BAL				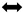	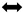	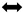		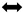	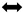	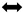					[97]
Never smokers: 9, mean age 36 years.Current smokers13, mean age 44 years*ND*	Acellular BAL					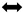	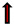		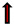			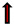	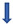	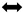		[102]
Never smokers: 11. Current smokers: 10. Mean age 36 years*LS*	Acellular BAL	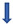				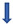 $										[100]
Never smokers: 20, mean age 32 years. Current smokers: 13,mean age 28 years.*LS*	Acellular BAL					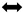			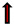			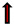				[103]
Never smokers: 12(aged 18 to 33 years).Current smokers: 8(aged 24 to 48 years).*LS and HS*	Acellular BAL				Phospholipids 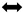		[98]
Never smokers: 65,mean age 28 ± 8 years.Current smokers: 23, mean age 53 ± 5 years. *LS*	Acellular BAL				Phospholipids 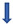		[101]
Never smokers: 14,mean age 54 yearsCurrent smokers: 20, mean age 42 years*LS*	Induced sputum			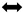	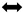	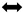	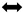	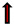	[99]

Smoking habit: ND = not defined; heavy smoker (HS) = more than 25 cigarettes a day; light smoker (LS) = fewer than 25 cigarettes a day [105]. BAL = bronchoalveolar lavage; FA = free fatty acids; GL = glycerolipids; Pa = phosphatidic acid; PC = phosphatidylcholine; PON-GPC = 1-palmitoyl-2-(9′-oxo-nonanoyl)-glycerophosphocholine; PG = phosphatidylglycerol; PE = phosphatidylethanolamine; PI = phosphatidylinositol; PS = phosphatidylserine; BMP = bis(monoacylglycerol)phosphate; PCe = ether phosphatidylcholine; Pep = plasmalogen phosphatydilethanolamine; APG = acyl phosphatydilglycerol; SM = sphingomyelin; Cer = ceramides; Chol = cholesterol; CE = cholesteryl esters. $ the total amount of PC was inversely correlated to the number of pack/year.

This is particularly important considering that the development or the pathological consequences of various smoking-related pulmonary diseases, such as COPD, could be related to alterations of the lipid fraction of the surfactant [24], even if it is unknown whether the dysregulation of surfactant is the cause or the consequence of the disease.

Studies to evaluate the effect of smoking on surfactant have been conducted also in experimental animal models (Table 2), principally mice, which have been shown to be a clinically relevant model system for some lung diseases, albeit with some limitations [106]. Whereas exposure to CS for a short period of time did not lead to any significant change in phosphatidylcholine levels [107], mice were more sensitive to prolonged treatment with CS. Indeed, a reduction in phosphatidylcholine levels has already been observed after a few months of exposure [108]. The changes in surfactant lipid composition were similar to those observed in COPD patients [24] but, to some extent, different from those observed in healthy smokers (Table 1).

**Table 2 ijms-24-11702-t002:** Dysregulation of different alveolar lipid types in animal models; controls vs. cigarette smoke exposed.

Organism;Type and Time of Exposure	Analysis	Tot. Lipids	Glycerophospholipids	Sphingolipids	Sterols	Ref.
	Pa	PC	PON-GPC	PG	APG	PE	PI	PS	BMP	PCe	PeP	Cer	Chol	
7 B6C3F1 mice;2 cigarettes/day, 5 days/wk, for 14 days	Acellular BAL				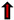											[109]
A/J mice: 16 control,16 CS exposed;filtered cigarette smoke for 4 or 8 wks	primary ATII total cell lysates															[108]
BALB/c C57BL/6 mice: 5 control,5 CS exposed;twice daily smoke from 12 cigarettes 3R4F, for 4 days	Acellular BAL			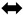												[107]
C57BL/6N mice;2 rounds of smoke of 50 min each, 5 consecutive days, for 12 wks	Acellular BAL	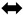												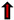		[92]
C57BL/6 mice:11 control,3 CS exposed;4 h of smoke from 20 3R4F cigarettes/6 months	Acellular BAL						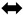					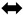				[44]

BAL = bronchoalveolar lavage; FFA = free fatty acids; GL = glycerolipids; Pa = phosphatidic acid; PC = phosphatidylcholine; PON-GPC = 1-palmitoyl-2-(9′-oxo-nonanoyl)-glycerophosphocholine; PG = phosphatidylglycerol, PE = phosphatidylethanolamine; PI = phosphatidylinositol; PS = phosphatidylserine; BMP = bis(monoacylglycerol)phosphate; PCe = ether phosphatidylcholine; Pep = plasmalogen phosphatidylethanolamine; Acyl PG = acyl phosphatydylglycerol; SM = sphingomyelin; Cer = ceramides; Chol = cholesterol; CE = cholesteryl esters.

Although, from a genetic point of view, it has been observed that exposure to CS leads to limited overlap between humans and mice, in a specific gene-by-gene comparison, the two species show the similar dysregulation of genes belonging to shared biosynthetic pathways, including genes associated with phospholipid metabolism/degradation pathways [110], suggesting that CS-exposed mice are particularly well suited to model the early phases of human COPD (GOLD stages 1 and 2) [111].

Differently from what was observed in mice, in the single experiment conducted on rats, CS chronic exposure—twice a day for 60 weeks in a nose-only exposure system—did not induce significant alterations in the phospholipid total content and profile, with the exception of desaturated PC. Exposure to tobacco smoke, however, significantly changed the surface activity of the pulmonary surfactant in adult rats [112].

In conclusion, even if animal models could be useful to study the effects of CS on the lipids of pulmonary surfactant, the results should be interpreted while keeping in mind the interspecific physiological differences of the respiratory system [113], which could lead to different effects and need to be taken into consideration when interpreting the results.

### 3.2. Molecular Mechanisms Leading to Cigarette-Smoke-Induced Modification of Surfactant Lipid Homeostasis

Numerous cellular mechanisms governing surfactant homeostasis in ATII cells [9] are impaired following exposure to CS and are reviewed in this section (Figure 2).

A recent multi-omics study conducted on mouse alveolar tissue was able to highlight the multiple interconnected molecular responses induced by exposure to CS: significant differential expression/abundance was found for 4324 mRNAs, 529 proteins, 173 metabolites, 43 miRNAs and 122 lipids [114]. The study, conducted on whole lung tissue, without differentiating between the numerous cell types present within the alveolar site, demonstrated that the tissue lipidomic profile changed extensively following exposure to CS, in agreement with a previous study [115]. Concerning specifically the activity of enzymes involved in surfactant metabolism, this study showed increased abundance of lysophosphatidylcholine acyltransferase 1 (LPCAT1), a multifunctional enzyme involved in the synthesis of saturated phosphatidylcholine (PC) via the remodeling of unsaturated PC (Lands cycle) in ATII cells [116], and reduced abundance of peroxiredoxin 6 (Prdx6), which acts as a calcium-independent lysosomal-type phospholipase A2 (PLA2) for surfactant lipids in MLBs, where the acidic internal pH, necessary for the basal PLA2 activity of the enzyme, is maintained [117].

The variations in the abundance of LPCAT1 and Prdx6 were previously shown in the same strain of mice [118], where a parallel increase in ATP-binding cassette class A3 (ABCA3), a transporter critically important for surfactant lipid secretion [119], and Slc34a2, a potential prognostic marker of oncological diseases [120], was also observed.

Recent research conducted on primary human ATII cells revealed that exposure to a CS extract (CSE) reduced *Lpcat1* gene expression [121]. These conflicting results in gene and protein expression may arise from biological aspects, e.g., post-transcriptional regulation with increased synthesis or, more likely, decreased degradation caused by proteasome impairments observed after CS exposure [122,123], or from confounding factors determined by the fact that these gene expression experiments were conducted on lung tissue as a whole, constituted of different cell types, or from the different responses to CS by organisms of different species (Table 1, Table 2, Table 3 and Table 4).

LPCAT1 is involved in several cellular functions. Besides having a biosynthetic role, LPCAT1 is involved in the transport of saturated PC from the endoplasmic reticulum to MLBs, interacting with transport proteins such as START domain containing 10 (StarD10) [132], and in the regulation of gene expression, after translocation into the nucleus and the induction of histone H4 palmitoylation [133]. The study of the effects of CS induced by its interference with this pathway certainly deserves further in-depth investigation, as a growing number of studies reveal key roles for the Lands pathway in specific mechanisms such as inflammation and immunity [116] and the involvement of LPCAT1 deficiency in promoting pulmonary emphysema via the apoptosis of alveolar epithelial cells [121].

The observed decrease in Prx6 levels is also intriguing, as prdx6^−/−^ mice chronically exposed (6 months) to CS showed increased pulmonary inflammatory responses, enhanced alveolar damage and airspace widening, increased compliance and decreased pulmonary resistance, which was not observed after acute exposure to CS [134]. Prdx6, besides having protective glutathione-dependent peroxidase activity against oxidative stress [135], has a crucial role in the remodeling pathway of DPPC and constitutes a key factor in surfactant turnover [136], because of its phospholipase and lysophosphatidylcholine acyl transferase activity [137]. Its reduced expression following chronic exposure to CS could induce interrelated effects of surfactant modifications, the activation of an inflammatory response and morpho-functional alterations of the lung, leading to emphysema, caused by a lack of protection from oxidative stress.

In mouse primary ATII cells exposed to CS for 4 or 8 months, an increase in PLA2 activity was observed, with a concomitant reduction in the PC level [108]. In this study, which did not investigate to which specific enzyme of the PLA2 superfamily this increase in activity is to be attributed [138], variations in the metabolism of cells exposed to CS were observed, with alterations in glycolysis and an increase in beta-oxidation and palmitate utilization. The latter was accompanied by an increase in the expression of the cellular and mitochondrial palmitate transporter CD36 and of carnitine-palmitoyl transferase [108].

An increase in PLA2 activity was also observed after the exposure of rat lung tissue slices to the tobacco-specific nitrosamines N-nitrosonornicotine (NNN) and 4-(methyl nitrosamino)-1-(3-pyridyl)-1-butanone (NNK) [139]. In this study, a significant increase in the mRNA and protein expression of two major cytosolic PLA2s, namely cPLA2 and sPLA2, was highlighted.

In contrast to what is known about the surfactant recycling pathway, no studies have been published to evaluate the variation in the activity of enzymes involved in the direct biosynthetic pathway (Kennedy pathway) of surfactant phospholipids. Considering the role of these enzymes in development, immunity and inflammation [116], an analysis of the effects of CS exposure on the direct biosynthetic pathway of surfactant phospholipids may be of value.

Besides a direct effect on the surfactant lipid metabolic pathways, indirect regulatory factors can also affect surfactant lipid metabolism following exposure to CS. For instance, CS-induced mitochondrial dysregulation can have consequences for surfactant lipid metabolism. Indeed, the exposure of murine lung epithelial MLE12 cells and of mouse primary alveolar epithelial cells to non-toxic CS concentrations induced an increase in the expression of mitochondrial fusion protein mitofusin 2 [140], in agreement with other studies showing the same effect in other lung cells [141,142,143]. Mitofusin 2 is an outer mitochondrial membrane protein that modulates lipid homeostasis by regulating the close apposition between ER and mitochondria, where enzymes catalyzing the synthesis of PS, PE and PC are located [144]. It has been recently demonstrated that mitofusins 1 and 2 have crucial importance in the production of surfactant lipids, regulating the synthesis of phospholipids and cholesterol in ATII cells [145].

The direct connection between changes in mitofusin 2 levels and the dysregulation of surfactant lipid production has not been investigated so far, but the topic could be intriguing, given the emerging central role of mitofusins in other types of CS-induced cell dysfunction, such as ER stress induction, the response to oxidative stress, cytoplasmic calcium alterations, cellular metabolism and surfactant production [146]. Indeed, mitofusins could represent a key connecting point, whose modifications following exposure to CS could impact cellular pathophysiology by linking different aspects of cellular function and the dysregulation of surfactant production.

Following synthesis, pulmonary surfactant lipids are stored in MLBs before secretion. The transport of surfactant lipids, such as PC, phosphatidylglycerol and cholesterol, within these organelles is mediated by the ABCA3 transporter, which is localized in the limiting membranes of MLBs [147]. The importance of this transporter in surfactant homeostasis is demonstrated by the fact that mutations in the ABCA3 gene can cause surfactant-deficient pulmonary disorders, such as fatal neonatal surfactant deficiency, chronic interstitial lung disease and diffuse parenchymal lung disease [148]. A recent study, conducted on rat lungs in vivo and on a human alveolar cell line in vitro, showed that CSE combined with LPS, a model widely used to study lung-inflammation-related diseases, can result in the downregulation of ABCA3, related to the PPARγ/P38 MAPK signaling pathway [149].

In both mice and humans, CS affected the pulmonary expression of the *abca1*, *abcg1*, *apoe* and *scarb1* genes, which codify for proteins involved in reverse lipid transport [150]. Deficiencies in these transport mechanisms could affect pulmonary surfactant homeostasis, as demonstrated for the ATP-binding cassette transporter A1 (ABCA1) and ABCG1, whose absence rapidly leads to lipid accumulation in alveolar macrophages and type II pneumocytes and to the accumulation of pulmonary surfactant related to pulmonary dysfunction [151].

The effect of CS on the other numerous lipid translocation proteins already identified in the MLBs [152] is unknown.

Surfactant secretion seems to be compromised also after exposure to CS. Indeed, CS inhibited the stimulated but not the basal release of radiolabeled PC in an in vitro system of adult rat ATII cells [153]. At the molecular level, an increase in mRNA expression for the purine receptor P2X4 was observed in human cells isolated from the BAL fluid and in the lung tissue of mice exposed to CS [154]. If and how this variation in P2X4, localized to the outer membranes of MLBs and involved in the facilitation of the secretion of pulmonary surfactant [152], could influence surfactant secretion upon exposure to CS has not been investigated.

P2X7 mRNA expression also appears to be increased after CS exposure [155]. P2X7, located in ATI cells, regulates lung surfactant secretion by ATII cells in a paracrine manner [156]. Although a correlation between an increase in P2X7 and the pathogenesis of CS-induced lung inflammation has been observed [155], the possible consequence for surfactant secretion is not known.

A few studies, conducted in vitro using synthetic surfactant substitutes, demonstrated that exposure to CS directly modified the surfactant’s chemical, mechanical and morphological properties, thus reducing its effectiveness in lowering surface tension in vitro [157,158]. More recently, it has been shown that oxidized lipids, such as 1-palmitoyl-2-(9′-oxo-nonanoyl)-glycerophosphocholine (PON-GPC), whose concentration increased after exposure to CS [109], reduced the stability of a surfactant monolayer model and its ability to resist high surface pressure. It has been suggested, but not proven, that the increase in oxidized lipids impairs the lipid cycle between the surface and the underlying lipid reservoirs, leading to serious consequences for respiratory dynamics [159,160]. These direct effects could be caused by the combined action of the numerous toxic compounds contained in CS. However, it was observed that even nicotine alone can interact, in a pH-dependent manner, with artificial pulmonary surfactant, impairing its surface activity [161].

Besides the direct effects on the rheological characteristics of the surfactant, the oxidation of phospholipids is a signal leading to immunomodulatory responses in different tissues and cells [162]. In addition to an increase in the levels of oxidized lipids such as PON-GPC [109], CS-induced oxidative stress led to an increase in cellular lipid peroxidation products—such as aldehydes 4-hydroxynonenal (HNE) and malondialdehyde (MDA)—in human alveolar A-549 cells [163].

Morissette and colleagues [107] have shown that the chemical dysregulation of surfactants, with increased oxidized lipids following cigarette smoking, could represent the triggering signal that leads to the accumulation of lipids in alveolar macrophages, giving them a foamy appearance. The accretion of lipids, in turn, induces the release of IL-1α, which plays an important role in maintaining pulmonary phospholipid homeostasis through a granulocyte macrophage colony-stimulating factor (GM-CSF)-dependent mechanism.

The same research group demonstrated that the activation of GM-CSF led to the recruitment of neutrophils from the circulation, which, shortly after exposure to CS, but not under physiological conditions, actively internalized surfactant phospholipids, supporting alveolar macrophages in restoring the disturbed surfactant homeostasis [25].

The critical role played by GM-CSF in the regulation of pulmonary surfactant homeostasis following exposure to CS is also suggested by the finding that autoantibodies directed against GM-CSF, or genetic mutations that disturb GM-CSF receptor signaling, lead to the development of pulmonary alveolar proteinosis (PAP). PAP is characterized by atypical surfactant accumulation, leading to respiratory failure and impairments in alveolar macrophage and neutrophil-mediated host defense [164].

The inflammatory response, which may represent the first defense mechanism against CS-induced damage, could be partly responsible for the observed effect on surfactant after exposure to CS. A comprehensive review analyzing the effect of smoking on surfactant/lipid processing by alveolar macrophages has been recently published [165].

## 4. Effect of Cigarette Smoke on the Protein Components of Surfactant

### 4.1. Cigarette Smoke Effect’s on Composition and Abundance of Surfactant Proteins

The effects of CS on the protein fraction of alveolar surfactant in humans and animal models are summarized in Table 3 and Table 4. As for the lipid fraction, to highlight the effects exclusively attributable to exposure to CS, we include only the studies conducted on healthy subjects and wild-type animal models. Furthermore, to ensure a complete overview of the effects, we include all the studies conducted since the 1970s.

The studies on humans have been all conducted analyzing the BAL fluid, which has the limitations illustrated previously (Section 3.1), except for two [54,127], in which endogenously generated droplets from breath aerosol (particles in exhaled air—PEx) were sampled. This method allows only the particles mainly generated in the lower airways and alveoli to be collected.

Most of the studies were conducted on SP-A and SP-D. CS caused an SP-D reduction in every study analyzed, except for one [126], in which the SP-D level remained unchanged. In the latter study, however, after LPS inhalation, the BAL SP-D level was lower in smokers than in non-smokers, suggesting that other coexistent external factors might have modified the effect of CS on SP-D [126]. The effect of CS on SP-A was more heterogeneous, as its level decreased in two studies [98,127], increased in one [54] and was unchanged in the last one [124].

As SP-A and SP-D play crucial roles in protecting the lung tissue from oxidative stress, inflammation and infection, many authors [81,98,125,126] hypothesize that their dysregulation in the smokers’ BAL might compromise the host defense functions of surfactant, increasing the susceptibility to respiratory infections and contributing to the development of lung diseases associated with CS. Conversely, another study [54] speculated that, as SP-A is known to promote toxin clearance, high SP-A levels in smokers could be a favorable response to respiratory toxins inhaled from tobacco smoke.

Besides inducing quantitative changes in surfactant proteins in BAL, smoke can also cause structural alterations [125]. Indeed, CS disrupted the quaternary structure of SP-D, leading to smaller subunits. This was probably caused by the reactive oxygen species formed in the smokers’ lungs.

Studies to evaluate the effect of exposure to CS on surfactant proteins were also conducted in experimental animal models, predominantly mice and rats (Table 4). As in humans, in animals, the most studied proteins are SP-A and SP-D. Unfortunately, even in animals, most studies analyzed proteins in the BAL fluid. In a study [128] in which the SP-A and SP-D levels were measured in both the BAL and a lung tissue extract, the results were conflicting: in BAL, both proteins were increased after exposure to CS, whereas in the lung extract, both were unchanged. Conversely, other studies found no difference in SP-D levels in BAL and lung lysate after exposure to CS [129,166].

CS’ effects on animal models appear to be the opposite to those in humans. Indeed, in most studies, exposure to CS increased the SP-D level. Only in two studies [166,167], the SP-D level was unchanged, and in one, it was decreased [130]. As observed in humans, SP-A varied more heterogeneously after CS exposure, decreasing in three studies, increasing in three others and remaining unchanged in one. Only two studies [114,131] analyzed all four surfactant proteins. One [114] showed that CS caused an increase in the levels of SP-A and SP-D, whereas those of SP-B and SP-C were unchanged; the other [131] showed a CS-induced reduction for every SP.

Many animal studies have shown a protective effect of SP-D against oxidative stress, inflammation, autophagy and apoptosis induced by CS. This has been demonstrated by administering recombinant SP-D [92,167], by studying the effects of smoking on SP-D-deficient mice [92,167] or by transfecting SP-D plasmid DNA into A-549 cells [168]. According to these studies, increased SP-D levels after exposure to CS would therefore be a positive and protective response against smoking-induced damage.

SP-D turnover also appeared to be altered in CS-exposed animals [91], with an increase in SP-D internalization by neutrophils. Surprisingly, the increased uptake was not accompanied by a decrease in BAL SP-A and SP-D levels, but rather by an increase.

Finally, the effect of CS was also tested on cultured alveolar epithelial cells [169,170,171] (Table 5). In A-549 cells, an epithelial cell line derived from human lung carcinoma, which has been widely used as an in vitro model of human alveolar type II epithelial cells [169], the mRNA expression of SP-B was increased after treatment with CS in a concentration-dependent manner, in agreement with another study where an increase in the mRNA expression of all four proteins was observed [170]. Differently, the mRNA and protein levels of SP-B were significantly decreased in the CS group compared with the control group in another study [168], where dexamethasone reduced the effect of CS. Finally, another study found a reduction in SP-C expression in human alveolar epithelial cells after exposure to CS and correlated this effect with apoptosis [171].

### 4.2. Molecular Mechanisms Leading to Cigarette-Smoke-Induced Modification of Surfactant Protein Homeostasis

The molecular pathways involved in the regulation of surfactant protein synthesis, intracellular trafficking and processing have been extensively characterized [172], but the effect of CS on these pathways remains largely undefined.

In a study conducted using comparative proteomic technology on lung tissue samples obtained from non-smokers and chronic cigarette smokers, an alteration in the levels of several proteins involved in protein synthesis/degradation was observed [173]. However, how the modified expression of these proteins may affect SP expression has not been analyzed here or elsewhere.

In the same study [173], an increase in the expression of the 14-3-3 protein sigma was observed. Several 14-3-3 isoforms can directly bind exon B of the 5′-untranslated region (5′-UTR) of human SP-A2 mRNA, regulating the transcription and translation of SP-A. Whereas most 14-3-3 isoforms are shown to regulate only SP-A2 expression, the 14-3-3 protein sigma affects the expression levels of both SP-A1 and SP-A2, suggesting the involvement of this hub protein in the general transcription of SP-A [174]. Furthermore, the supramolecular complex comprising the 14-3-3 delta that interacts with the 5′UTR of SP-A2 includes several proteins, such as ribosomal, cytoskeletal and translation factor proteins (i.e., calreticulin, tubulin, actin and 60S ribosomal proteins) [175], that are altered following exposure to CS [173]. However, the direct correlation between exposure to CS and the 14-3-3 supramolecular complex-mediated alteration of SP-A has not been investigated.

Finally, S100-A9/calgranulin C, a protein with inflammatory activity, was found to be downregulated in the lungs of chronic smokers [173]. Among other targets, S100-A9 binds the transcription factors NF-kB and P53, which modulate SP-A [176] and SP-C expression [177]. NF-κB DNA-binding motifs were found in the promoter regions of the *spa* [178] and *spb* [179] genes by bioinformatic searches. The downregulation of S100-A9 suggests the possible dysregulation of the NF-kB-mediated transcription of several SPs associated with an anti-inflammatory response in smokers.

Surfactant protein homeostasis is regulated by other gene transcription and/or mRNA stability regulatory pathways. Two main transcriptional factors implicated in SP expression are thyroid transcription factor-1 (TTF-1) and nuclear factor 1 (NF1) [180]. Unfortunately, to our knowledge, no study has evaluated the effects of CS on these factors. Since it has been shown that TFF-1 oxidation reduces its DNA-binding affinity [180], it would be interesting to study CS’ effect on the regulatory function of this transcriptional factor.

As SP-D gene expression has been shown to be positively regulated at the transcriptional level by AP-1 (a heterodimer of the fos and jun protein kinase families) and by CCAAT/enhancer binding protein-β (C/EBPβ) [181,182], it was investigated [129] whether the increased level of SP-D found in mice lungs exposed to CS was related to the increased expression of these factors, measuring the expression of the *Fra-1*, *junB*, *junD*, *c-fos* and *C/EBPβ* mRNAs by semiquantitative RT-PCR in lung tissue homogenates. The gene expression of *Fra-1*, *junB* and *C/EBPβ* was significantly upregulated in the lungs of CS-exposed mice, whereas *c-fos* gene expression was significantly attenuated. *JunD* gene expression tended to increase after exposure to CS but without statistical significance. Upregulation of the DNA-binding activity of AP-1, partially dependent on *Fra-1*, *junB* and *junD*, was also found in the lungs of CS-exposed mice. The involvement of AP-1 in the regulation of SP gene transcription has also been demonstrated for SP-A and SP-B, although the effects of CS have not been evaluated [183].

Hydrophobic SP-B and SP-C undergo extensive post-translational processing in the lysosomal compartment by intracellular proteases such as napsin A, cathepsin H and pepsinogen C [70]. The activation of Notch receptor 1 (Notch1) in vitro and in vivo was found to decrease napsin A and cathepsin H activity, with consequent abnormal SP-B and SP-C processing [184]. This effect has been proposed to be mediated by the activator of transcription–Janus kinase (Jak–Stat) signaling pathway. Jak–Stat has been shown to activate SP-B mRNA expression in H441 cells [185] and to regulate SP-B secretion, enhancing ABCA3 expression [186]. Although, in these works, the direct CS effect on napsin A and cathepsin H activity has not been studied so far, CS has been shown to dysregulate the *Notch* gene expression [187,188], suggesting that the effects of CS on SP-B and SP-C levels could depend also on this mechanism.

Concerning the hydrophilic SP-A and SP-D, their synthesis and processing, including their glycosylation and oligomerization, occur at the endoplasmic reticula of type II cells [9]. The mechanisms of their post-translational processing and secretion are rarely studied, and the effects of smoking on these mechanisms are considered even more rarely.

SP-B and SP-C are secreted through MLBs together with lipids. For this reason, every factor that influences the secretion and removal of lipids from the alveolar fluid can also influence the turnover of these two proteins (see Section 3.2).

The innate immune functions of SP-A and SP-D are strictly dependent on their structures [84,166]. The aldehydes present in CS, such as acetaldehyde, acrolein and formaldehyde, can affect SP-D and SP-A’s structures at different levels. CS has been shown to change the quaternary structure of SP-D from a multimer to trimer and monomer, leading to a decrease in its antimicrobial function [125,166,189]. Moreover, CS induced acrolein modification in the carbohydrate-binding domains of both SP-A [84] and SP-D [166], interfering with their ability to bind allergens and pathogens and to induce macrophage phagocytosis. The immune defects mediated by CS can lead to the development of lung diseases such as COPD, respiratory infections and lung cancer.

Finally, although the analysis of these articles is beyond the scope of this review, two studies investigated the molecular mechanisms underlying the protective effect of SP-D against the adverse effects of CS [92,167].

The cellular mechanisms governing SP homeostasis in ATII cells and impaired following exposure to CS are summarized in Figure 3.

## 5. Molecular Impact of Electronic Cigarette Vapor on Pulmonary Surfactant

An electronic cigarette (e-cigarette) is an apparatus that allows the aerosolization of liquids (e-liquids) generally consisting of three components: a humectant, such as propylene glycol (PG) or vegetable glycerin (VG); a drug, such as nicotine, tetrahydrocannabinol or cannabidiol; and a flavoring chemical. In between electronic and traditional cigarettes are “heat not burn” products, electronic devices that, unlike e-cigarettes, contain tobacco, which is heated to a high temperature, without burning [114]. For this reason, companies claim that it has fewer chemicals than burned tobacco.

Since their introduction, the popularity of e-cigarettes has increased significantly, especially among adolescents and young adults [190]. E-cigarettes were initially intended as a smoking cessation aid and so were perceived as a safer alternative to traditional cigarettes, although they also expose users to many toxins and carcinogens. Whereas, because of their recent introduction, few data are available on the long-term effects of e-cigarettes, the reports of e-cigarette or vaping product use associated lung injury (EVALI) have increased dramatically in the last few years [191]. EVALI is characterized by dyspnea, fever, leukocytosis and bilateral ground-glass opacity and demonstrates symptomatic, functional, and radiological improvement following steroid administration [192]. There is strong circumstantial evidence that links the addition of vitamin E acetate with the occurrence of EVALI [193,194]. Nevertheless, other toxins in vaping solutions cannot be excluded. Indeed, potentially harmful constituents present in e-cigarette vapor include carbonyl compounds, volatile organic compounds, tobacco-specific nitrosamines and heavy metals [195]. Moreover, it has been demonstrated that the generation of the aerosol results in the formation of a large quantity of new products, including formaldehyde and acetaldehyde [196]. The study of the adverse effects of e-cigarettes is complicated by the multitude of vaping liquids and devices. Indeed, consumers can personalize their vaping experience by adding flavors or drugs to the liquid base; moreover, the market offers devices with different values of voltage and resistance [196].

Most studies investigating the effect of e-cigarettes on alveolar surfactant evaluate the influence of e-cigarette liquids (direct liquid, vapor condensate and vapor), single liquid components such as PG and VG or additives such as vitamin E acetate and cannabinoids, on model surfactant films constituted by an appropriate phospholipid mixture [197,198,199,200] or by animal lung surfactant extract [196,201,202]. The nature of the lipid film used, the component of the e-cigarette tested, the method utilized and the results of these studies are summarized in Table 6. Most of these studies have shown an effect of e-cigarettes on the mechanical properties of the surfactant, negatively impacting lipid packing, film stability and the turnover of the phospholipids from the air–liquid interface and the underlying multilayer reservoir. These studies show that some of the e-cigarettes’ detrimental effects can be explained by a physical interaction between the components of the e-liquids and the surfactant.

Because of the recent introduction of e-cigarettes on the market, studies on the cellular effects of these devices are limited, especially the long-term ones. However, it is quite evident that these devices also have effects on a cellular level, as described in a recent and comprehensive review [194]. Unfortunately, as regards the effects on surfactant, experimental works in the literature are very limited. Only two studies have investigated the effect of exposure to e-cigarette vapor in vivo [203,204]. The first one studied chronic inhalation exposure to e-cigarette aerosols derived from only a vehicle (PG/VG) or a vehicle plus nicotine in a murine model. Exposure to e-cigarette aerosols alters the lipids in lung macrophages and ATII cells. Macrophages isolated from the BAL of mice exposed to e-cigarette aerosols showed increased lipid accumulation independent of nicotine. The increased concentration of intracellular lipids was not accompanied by an increase in intracellular glycerol, indicating that the source of accumulated lipids might have been endogenous, rather than exogenous. The increase in lipid concentration was attributable to an increase in cellular phospholipids, with the enrichment of desaturated phospholipids and cholesterol esters, whereas other lipids (e.g., triglycerides) showed no significant alterations. The analysis of the extracellular lipids in BAL showed the enrichment of the lipid species that are most prominent in surfactant.

In the same study, in ATII cells isolated from mice exposed to e-cigarette aerosols, the total number of MLBs was unchanged, but the number of poorly organized, irregular MLBs was increased. Furthermore, the expression of *Abca3* in BAL cells was upregulated in response to e-cigarette exposure, with the vehicle plus nicotine demonstrating an additive effect, whereas the expression of *Abca1* and *Abcg1* in lung homogenates was reduced. In the lung homogenates with e-cigarette exposure, upregulated gene expression also of the phosphatidylcholine-modifying enzymes *Lpcat1* and *Pcyt1a* was observed. Mice treated with e-cigarette aerosols showed significantly reduced SP-D concentrations in BAL fluid and significantly reduced mRNA transcripts of the *Sfptd* and *Sfpta* genes in lung homogenates, whereas no significant changes were found in *Sfptb* and *Sfptc*.

In the second study [205], mice were treated with vapor from an e-cigarette liquid composed of 70% PG/30% VG, with or without vanilla flavoring. The levels of SPs were analyzed in lung homogenates, finding no effect on the gene expression of *Sfpa*, *Sfpb*, *Sfpc* or *Sfpd*. Conversely, the PG/VG e-cigarette aerosol dysregulated genes related to both phospholipid and lipid homeostasis (Il-6, F2, Snca and Aldh8a1). E-cigarette emissions contain many toxic chemicals that are also found in traditional cigarettes. Furthermore, e-cigarettes, as with traditional ones, have also been shown to induce oxidative injury and chronic inflammation. How and if these effects can induce alterations in surfactant homeostasis in analogy with what is observed for cigarette smoke offers a starting point that could allow us to increase the limited knowledge that unfortunately is available currently on this topic.

## 6. Surfactant-Related Biomarkers Altered in Plasma by Cigarette Smoke or Electronic Cigarette Vapor

Cigarette smoking, as well as the use of electronic cigarettes, induces changes in numerous plasma metabolites. Many putative circulating biomarkers [205,206] correlated with an increased risk of developing pulmonary and cardiac diseases have been reported in the plasma of both smokers and e-cigarette users [205,206]. For instance, CS is known to induce sphingolipid dysregulation in smokers’ lungs and surfactant (Table 2). Some articles have described dysregulated plasma sphingolipids associated with lung cancer and chronic obstructive pulmonary disease (COPD) phenotypes [207,208]. A recent article highlights an increase in plasma sphingosine levels in the serum of cigarette smokers but not in e-cigarette users, suggesting that the induced increase in cardiovascular risk is mediated by a number of other different factors, which in any case appear to be dysregulated in the plasma of e-cigarette users [205].

Many studies have measured the blood levels of SPs whose amounts in serum are negligible in physiological conditions [73,209] but increase considerably (five-fold higher than the basal value for SP-B) in smokers [69,73,84,210,211,212].

SPs are mainly synthesized in the lungs and secreted into the alveolar space [53,54,81,90,213], although they have been detected in a variety of extrapulmonary tissues and fluids [53,58,59,60]. Thus, it is generally assumed that the origin of SPs in the blood is mostly their drainage from the lung into the vascular compartment and that their augmentation in smokers’ serum is consequent to increased lung leakage into the bloodstream [73,90].

The routes and the exact mechanisms by which SPs enter the circulation have long been debated, and it is now accepted that increased transcytosis, basolateral secretion and lymphatic clearance from the interstitium are not responsible for the increases in SP serum concentrations [73,214,215,216]. Conversely, intravascular leakage and the circulating levels of SPs increase in conditions characterized by pulmonary inflammation and/or lung injury [63,90]. Moreover, CS leads to lung epithelial and endothelial damage, increasing alveolar–capillary membrane permeability [126] and pulmonary leakage; these CS-induced effects occur both acutely, through the release of vasoactive mediators, and chronically, through the destruction of lung parenchyma integrity [73]. Because of the increased leakage of SPs towards the vascular compartment, the presence of these proteins in the alveolar fluid, BAL and PEx would be reduced [10,25,73,126,127,166]. Since SPs leak into the circulation when pulmonary integrity fails, pneumoproteinemia (high serum SP levels) has been suggested as a non-invasive peripheral marker of CS-induced loss of lung epithelial barrier integrity and alveolar injury [10,73,166]. This could be very useful for early diagnosis, as the small airways are difficult to access and their pathological changes, asymptomatic in the early stage, are observed only in the subsequent evolution of lung disease. Serum SP increases may instead reflect early alveolar inflammatory responses to CS, detectable even when lung function is within normal limits [54,73]. Moreover, at present, there is still a lack of markers sensitive enough to identify early-stage alveolar pathological alterations before damage to the small airways becomes evident and permanent [54,73].

A study evaluating SP-A and SP-B as early markers of alveolar damage in smokers shows that SP-B appears to be a more sensitive indicator of lung injury [73], probably in relation to its smaller size (~18 vs. ~650 kDa) and the consequently greater ability to easily cross the alveolar–capillary membrane [68,69,73,210].

It has been suggested that serum SP levels could also have a prognostic purpose, as they have been considered biomarkers of lung injury severity and predictors of unfavorable outcomes [10,213].

In smokers’ serum, the SP-B and SP-A levels are influenced by exposure to CS as they positively correlate both with pack/years and with the number of cigarettes smoked daily [69,73]. Moreover, serum SP-B (more than SP-A) is inversely related to FEV1/vital capacity, suggesting a correlation between obstructive disease and alveolar leakage, whereas this correlation is lacking in non-smokers [73]. Finally, in COPD patients’ serum, SP-A was significantly higher in exacerbated than in stable (free of exacerbation) subjects [217].

SP-D serum levels also have shown a positive correlation with exposure to CS (pack/year consumption) and a negative one with lung function parameters, whereas, in COPD patients, they are further augmented in the case of exacerbation and are linked to increased mortality [92,212]. Recent studies indicate that SP-D can be a predictive marker of COVID-19 disease and its outcomes [218,219,220,221,222,223].

Several studies have shown that the serum expression level of pro-surfactant protein B (pro-SFTPB) is increased in patients with lung cancer, with a poor prognosis and shorter survival, thus indicating a diagnostic and prognostic role of serum pro-SFTPB in detecting human cancer [224,225,226,227]. Indeed, lung cancer cells show dysregulated SFTPB synthesis with the overexpression of pro-SFTPB and an inability to post-translationally modify the precursor into the mature SP-B form. Serum pro-SFTPB and SP-D expression levels were higher in HIV-positive patients [228], suggesting that dysregulated serum pro-SFTPB expression levels may be associated with other human diseases.

## 7. Conclusions

Alveolar surfactant, a lipo-protein complex covering the epithelial lining fluid of the alveoli, is essential for respiratory dynamics and for pulmonary immunoprotection.

CS is known to seriously affect the physiology of the lungs and it is related to the development of smoking-related diseases with increased morbidity and mortality.

In addition to cytopathological effects, both on alveolar and other cell types—such as macrophages and neutrophils—of pulmonary alveoli, CS also affects surfactant at different levels. E-cigarettes have been shown to deleteriously impact also the surfactant system. However, research on this topic is still at an early stage.

Several articles—cited in Section 3.1 and Section 4.1 and summarized in the tables—analyze the effect of CS on pulmonary surfactant in humans (Table 1 and Table 3) and in animals used as an experimental model system (Table 2 and Table 4).

The general picture that emerges from the analysis of the literature shows great heterogeneity in the response to CS exposure on both the lipid and protein surfactant components.

This heterogeneity in the results is attributable to the different experimental procedures implemented for the analysis of the surfactant, to the different degrees of exposure to smoke and to the different characteristics of the subjects under examination (generally limited to a few dozen) and highlights the need for further investigations.

Additional studies should be conducted on larger and more homogeneous populations, taking into account factors such as age, smoking habits and previous exposure to deleterious environmental factors, which can influence surfactant homeostasis by themselves.

Although animal models are suitable for the modeling of human lung diseases, such as COPD [111], the heterogeneity in responses in animals compared to that observed in humans should be highlighted considering the interspecific physiological differences in the respiratory system [113].

At the molecular level, it has been demonstrated that CS promotes the direct oxidation of lipids, altering surfactant properties and the dynamics of the lipid monolayer and underlying membranous structures during the respiratory cycle.

Furthermore, several studies highlight the fact that CS can alter the expression and activity of proteins involved in the synthesis, transport, secretion and reabsorption of lipids and surfactant proteins, mediated by both ATII cells and alveolar macrophages.

Extensive studies have been conducted on the role of macrophages in surfactant-reuptake-related pathophysiological processes, which have only been partially addressed in this review, as they have been recently and exhaustively presented by others [165].

Most of the studies (reported in Section 3.2 and Section 4.2) aimed at analyzing the molecular impact of CS on the mechanisms mediating surfactant homeostasis in alveolar epithelial cells are limited to observational evaluations, whereas an analysis of the molecular correlations that induce or follow these variations is lacking. Although, for example, it has been ascertained that CS alters the expression of enzymes involved in the metabolism of surfactants, such as LPCAT and Prdx6, nothing is reported regarding the molecular pathways underlying this variation. Although it has been proven that there is an effect of CS on factors, such as mitofusins or 14-3-3 protein, that can indirectly influence surfactant homeostasis, the molecular correlations of these changes can be only deduced from a literature review and have yet to be analyzed experimentally. Furthermore, the signal transduction pathways leading to the variations in gene transcription underlying the expression of enzymes involved in lipid or SP synthesis after CS exposure remain to be elucidated.

Concerning e-cigarettes, several studies have shown a direct effect of some e-liquid components on the mechanical properties of the surfactant. On the other hand, very little is known about their cellular effects on surfactant homeostasis, and even less about their molecular mechanisms. This topic still needs to be investigated, particularly considering that these devices are gaining popularity, especially among adolescents and young adults.

A thorough analysis of the effects of CS, made easier nowadays thanks to the ever-increasing development of -omics methods, is essential to identify the initial molecular effects promoting the pathophysiological alterations that lead to the development of several lung diseases. However, these up-to-date methodologies, which have and will permit us to obtain a broad view of the simultaneous alterations induced by CS, must be paralleled by comprehensive cellular physiopathological analyses to consolidate the experimental observations and understand the CS-related causes and consequences at the molecular and cellular levels.

Even if the best approach to preventing CS damage remains the cessation of smoking, these studies will allow us to develop not only preventive strategies but also more targeted personalized therapies in the case of damage induced by CS.

## Figures and Tables

**Figure 1 ijms-24-11702-f001:**
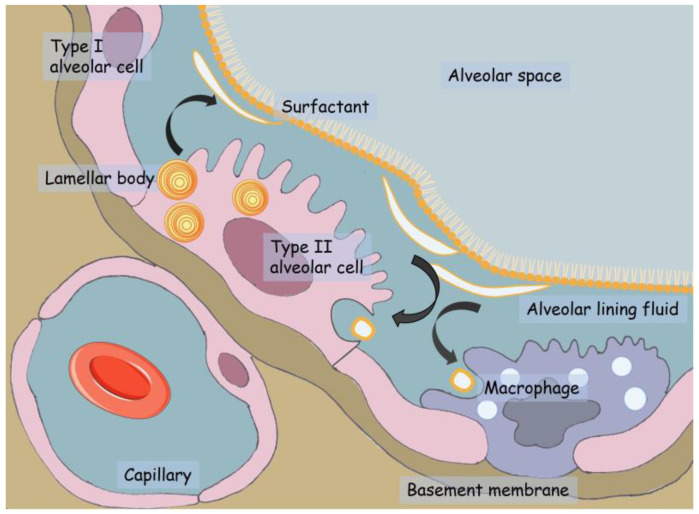
Schematic representation of the alveolar region.

**Figure 2 ijms-24-11702-f002:**
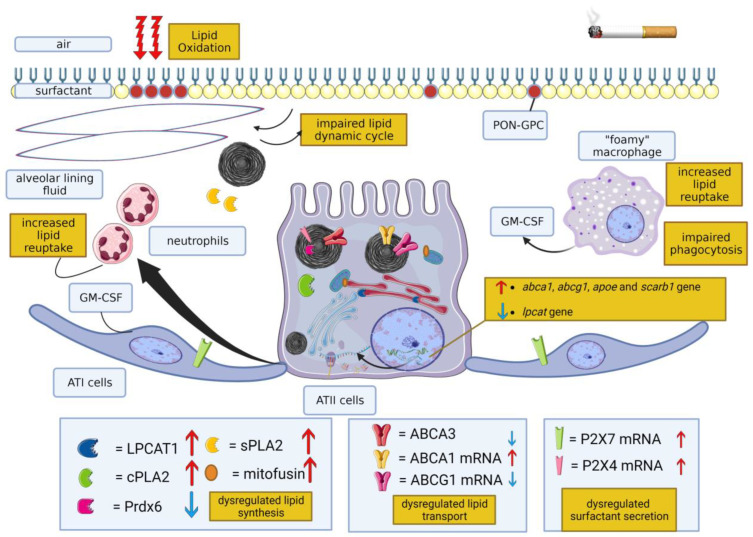
Schematic representation of cellular and molecular impacts of CS on surfactant lipid homeostasis. Besides the oxidation of the lipids constituting the surfactant, with alterations in the functionality of the surfactant and the dynamic lipid cycle between the surfactant and the underlying lipid reservoirs, CS causes the dysregulation of various enzymes involved in the synthesis, transport, MLB accumulation and secretion of surfactant lipids. Parts of the figure were drawn using pictures from Servier Medical Art. Servier Medical Art by Servier is licensed under a Creative Commons Attribution 3.0 Unported License (https://creativecommons.org/licenses/by/3.0/, accessed on 19 June 2023).

**Figure 3 ijms-24-11702-f003:**
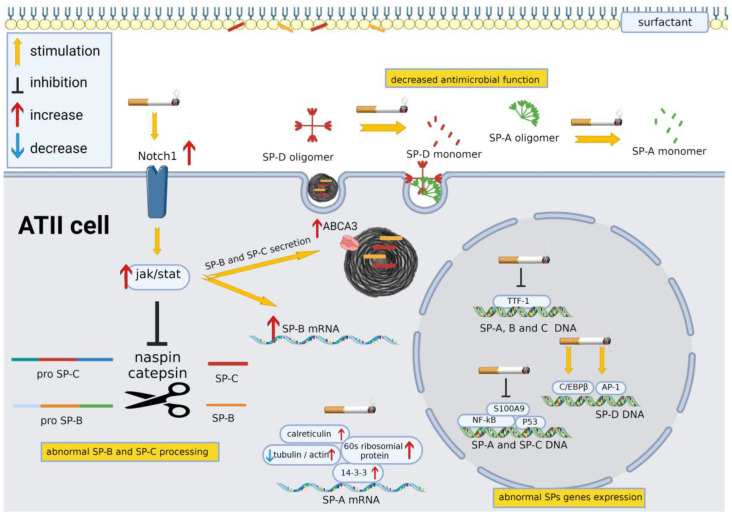
Schematic representation of cellular and molecular impacts of CS on surfactant protein homeostasis. CS causes the dysregulation of surfactant protein gene expression, abnormal SP-B and SP-C processing and a reduction in SP-A and SP-D’s antimicrobial activity. Parts of the figure were drawn using pictures from Servier Medical Art. Servier Medical Art by Servier is licensed under a Creative Commons Attribution 3.0 Unported License (https://creativecommons.org/licenses/by/3.0/, accessed on 19 June 2023).

**Table 3 ijms-24-11702-t003:** Dysregulation of different surfactant proteins in smokers vs. non-smokers.

Subjects	Smoking Habit	Analysis	SP-A	SP-B	SP-C	SP-D	Ref.
Never smokers: 12 (aged 18 to 33 years).Current smokers: 8 (aged 24 to 48 years).	LS and HS	Acellular BAL					[98]
Never smokers: 22 (15 young, 7 elderly).Current smokers: 82 (20 young, 62 elderly).	ND	Acellular BL and BAL	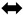				[124]
Never smokers: 10 (21–36 years), young smokers: 10 (21–49 years), elderly smokers: 20 (40–75 years), elderly smokers with COPD: 20 (40–75 years).	ND	Acellular BAL					[125]
Never smokers: 10,mean age 40 years.Current smokers: 10,mean age 40 years	ND	Acellular BAL				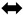	[126]
Never smokers: 29,mean age 52 years.Current smokers: 37,mean age 59 years	LS	PEx	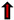				[54]
Never smokers: 97,age 42–67 years.Current smokers: 12,age 42–67 years	ND	PEx					[127]

Smoking habit: (ND) = not defined; heavy smoker (HS) = more than 25 cigarettes a day, light smoker (LS) = fewer than 25 cigarettes a day [105]. BAL = bronchoalveolar lavage; BL = bronchial lavage; PEx = particles in exhaled air.

**Table 4 ijms-24-11702-t004:** Dysregulation of different alveolar surfactant proteins in animal models; controls vs. cigarette smoke exposed.

Organism	Type and Time of Exposure	Analysis	SP-A	SP-B	SP-C	SP-D	Ref.
Female Sprague-Dawley rats, 10 controls (5 not treated, 5 sham-treated), 5 CS exposed.	2 cigarettes/day, 7 days/week, for 70 weeks	Acellular BALTotal mRNA and proteins from lung tissue	 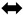	 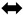			[128]
Male B6C3F1 mice,20 controls, 20 CS exposed.	2 cigarettes/day, 5 days/week, for 6 months	Acellular BAL and total mRNA from lung tissue				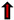	[129]
Male C57BL/6N wild type (WT)n = 5–7	2 rounds of exposure to CS of 50 min each, with a smoke-free interval of 30 min, 5 day/week for 12 weeks or for 3 days	Acellular BAL				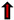	[92]
Female BALB/c mice, 5 controls, 5 CS exposed	2 h every morning for 4 days to the mainstream CS of 24 cigarettes with filters removed	Acellular BAL	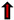			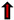	[91]
Male Wistar rats, 10 controls, 10 CS exposed	2 cigarettes/day for 45 days	Total proteins of lung tissue					[130]
C57BL/6J mice, 4 controls, 4 CS exposed	30 µL/day of CSE administered via oropharyngeal instillation for 4 weeks	Total mRNA of lung tissue					[131]

BAL = bronchoalveolar lavage; CS = cigarette smoke; CSE = cigarette smoking extract.

**Table 5 ijms-24-11702-t005:** Dysregulation of different alveolar surfactant proteins in cultured cells; control vs. cigarette smoke exposed.

Cell Line	Type and Time of Exposure	Analysis	SP-A	SP-B	SP-C	SP-D	Ref.
A-549	30 mg/mL CSE for 48 h	Total mRNA		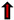			[169]
A-549	Tobacco extract (100 μg/mL) for 1 h	Total mRNA	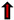	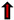	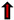	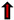	[170]
Primary HAEC	2.5% CSE for 48 h	Total mRNA					[171]
A-549	10% CSE for 24 h	Total mRNA and proteins					[168]

CSE = cigarette smoke extract; HAEC = human alveolar epithelial cells.

**Table 6 ijms-24-11702-t006:** Influence of e-cigarette liquids on model surfactant films.

Nature of Lipid Film	Component Tested	Method	Results	Ref.
Mixture of DPPC, POPG and PA	E-cigarette vapor	Langmuir trough simulating the alveolar environment	Reduced compressibility of surfactant film and impaired capacity to reduce the surface tension.	[197]
Calf lung surfactant extract	Different volumes of cigarette smoke or e-cigarette vapor bubbled in the subphase	Langmuir trough and atomic force microscopy	Both e-cigarette vapor and conventional cigarette smoke affect surfactant lateral structure; only cigarette smoke disrupts surfactant interfacial properties.	[201]
Calf lung surfactant extract	PG and VG at different proportions and concentrations added to the surfactant	Dynamic surface tension measured under a simulated breathing cycle using drop shape method	Only e-liquid concentrations > 200× higher than the estimated average dose after a single puffing session induced measurable changes in surfactant biophysical activity and only ultra-high concentrations inactivated the surfactant.	[202]
Pure DPPC or lipid mixture (DPPC/POPC/POPG/Chol)	Vitamin E acetate added to the lipid mixture	Neutron spin echo spectroscopy	Vitamin E acetate increased phospholipids membrane fluidity and compressibility.	[198]
Pure DPPC	Aerosol of liquid composed by PG and VG	Attenuated total reflectance-Fourier-transform infrared spectroscopy	PG and VG modified the molecular alignment of the DPPC surfactant, affecting the surface tension at the air-water interface.	[199]
DPPC and POPG mixture	5 mol% of vaping additives (vitamins and/or cannabinoids) added to the DPPC and POPG mixture	Langmuir-Blodgett trough and Brewster angle microscopy	Vaping additives negatively impact lipid packing and film stability, induce material loss upon cycling and significantly reduce functionally relevant lipid domains.	[200]
Bovine lipid extract surfactant	Vapor of a mixture of PG and VG at different ratios and with different devices with or without flavoring and nicotine	Constrained drop surfactometer	Minimum surface tension increased significantly after exposure to the e-cigarette aerosol. Variations in device used, addition of nicotine and temperature of the aerosol had no additional effect. Two e-liquid flavors, menthol and red wedding, had further detrimental effects.	[196]

## Data Availability

Data sharing not applicable.

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
