# Peer review of "Molecular Impact of Conventional and Electronic Cigarettes on Pulmonary Surfactant"

_ijms, 2023, doi:10.3390/ijms241411702_

Round 1

Reviewer 1 Report

The review contains a decent array of studies and extensive results in relation to cigarette exposure and effect upon pulmonary surfactant. The overall flow of the piece is however diminished as some sections are disjointed. The articulateness can also be improved for several sentences. Please modify several sentences in which the formality is diminished.

Diagrams are well constructed, informative and easy to follow.

Reorganise the abstract for succinctness: Eg.  “Tiny aqueous hypophase…” – remove tiny
“Qualitative and quantitative effects…” – remove qualitative and quantitative

Line 38 – “Anyway, ….” Edit sentences with this informal opening  

COPD can be defined and summarised before later being mentioned in the article.

The source of surfactant can be identified (Clara cells) and expanded upon in the introduction.

The writing in the electronic cigarette section is a touch speculative – focus on what is known, what is unknown, and what further research can be done.

Even the use of e-cigarettes, advertised as a healthier alternative to CS, turns out to

have a deleterious impact on the surfactant system. However, investigations in this area are

still in the early stages. reword to be more succinct and adequately articulated

Some studies have linked the addition of vitamin E acetate with the

occurrence of EVALI; anyway, there is not enough evidence that vitamin E acetate is the

sole cause of injury - this needs to be further explored – the CDC (US) declared that vitamin E acetate addition was the strong likely cause for the 2019 EVALI epidemic in USA.

From the analysis of the available literature, it remains evident that the effect of CS on 856

the various lipid and protein components and, above all, its molecular impact remains

largely to be investigated. - is this conclusion rigorous? – start with the conclusion of existing studies and then move onto the statement that the molecular impact remains largely to be investigated.

The overall flow of the piece is however diminished as some sections are disjointed. The articulateness can also be improved for several sentences. Please modify several sentences in which the formality is diminished.

Edit sentences with informal openings  

Reviewer 2 Report

This is a well written and well organized manuscript discussed the cigarette smoke and e-cig exposure induced dysregulation in lung surfactant and associated lun disease and injury process. The manuscript looks fine, only some minor suggestions can be considered to organize a bit of the review article to improve the structure of logic flow. 

1. in the sub-section 2.1, discussed about the different lung surfactant component. It is better as an individual section as background introduction.

2. section 3, it directly jumped to section 5, please correct accordingly.

3. Section 2 and Section 3 are both cigarette smoke while limited sections discussed about e-cig related dysregulation of lipid surfactant homeostasis. Please consider discuss a bit more of how e-cig vaping associated with surfactant dysregulation, and it's molecular mechanism.

4. Author discussed about sphingosine, please refer to this ref and discuss the increased sphingosine-related metabolites from smokers' serum while not in E-cig user (PMID: 34072305)
